# The Role of Emotional Regulation and Affective Balance on Health Perception in Cardiovascular Disease Patients According to Sex Differences

**DOI:** 10.3390/jcm9103165

**Published:** 2020-09-30

**Authors:** Bárbara Luque, Rosario Castillo-Mayén, Esther Cuadrado, Tamara Gutiérrez-Domingo, Sebastián J. Rubio, Alicia Arenas, Javier Delgado-Lista, Pablo Pérez Martínez, Carmen Tabernero

**Affiliations:** 1Maimonides Biomedical Research Institute of Córdoba (IMIBIC), 14004 Córdoba, Spain; esther.cuadrado@uco.es (E.C.); tamara.gutierrez@uco.es (T.G.-D.); sjrubio@uco.es (S.J.R.); aarenas@us.es (A.A.); delgadolista@gmail.com (J.D.-L.); pabloperez@uco.es (P.P.M.); 2Department of Psychology, University of Córdoba, 14071 Córdoba, Spain; 3Department of Didactics of Experimental Sciences, University of Córdoba, 14071 Cordoba, Spain; 4Department of Social Psychology, University of Seville, 41018 Seville, Spain; 5Lipids and Atherosclerosis Unit, Department of Internal Medicine, Reina Sofía University Hospital, 14004 Córdoba, Spain; 6Department of Medicine (Medicine, Dermatology and Otorhinolaryngology), University of Córdoba, 14004 Cordoba, Spain; 7CIBER Fisiopatología de la Obesidad y Nutrición (CIBEROBN), Instituto de Salud Carlos III (ISCIII), 28007 Madrid, Spain; 8Instituto de Neurociencias de Castilla y León (INCYL), University of Salamanca, 37007 Salamanca, Spain

**Keywords:** Gender, women’s health, emotional regulation, cardiovascular health

## Abstract

One of the challenges of aging is the increase of people with chronic diseases, such as cardiovascular disease (CVD). Men and women experience the disease differently. Therefore, it has an impact on how CVD is treated and its outcomes. This research analyzed the relationship between psychosocial variables and health promotion among cardiovascular patients, paying special attention to sex differences. A longitudinal study with cardiovascular patients (747 in phase 1 (122 women) and 586 in phase 2 (83 women)) was carried out. Participants were evaluated based on their sociodemographic characteristics, affective balance, regulatory negative affect self-efficacy, stress and anxiety regulation strategies, and perceived global health. Results showed that men presented significantly higher scores in positive affect, affective balance, and self-efficacy to regulate negative emotions, while women presented significantly higher scores in negative affect and the use of passive strategies to cope with stressful situations. Regression analyses showed that all psychological variables studied in phase 1 were significant predictors of health perception in phase 2. According to the results, it is necessary to include strategies to improve cardiovascular health through education and emotional regulation, with a gender focus.

## 1. Introduction

Cardiovascular disease (CVD) is the leading cause of death worldwide [1]. In Europe, CVD causes around four million deaths each year, which represents 45% of all deaths [2]. Considering sex differences, previous studies have shown that women are less likely to experience acute coronary syndromes, but they have a worse evolution following a cardiovascular incident compared to men [3,4,5]. The risk of CVD in women is often underestimated due to the misperception that women are more “protected” than men against CVD [6]. The clinical manifestations of CVD also differ between men and women [3]. Among women, CVD tends to appear 7–10 years later than in men and is the main cause of death in women above 65 years of age [7,8]. The fact that men and women experience the disease differently may have had an impact on how CVD is treated and its outcomes. Differences have also been found among symptoms [9], with it being more likely for women not to report chest pain, to present with fewer changes in the electrocardiogram, and to have high troponin levels. Generally, it can be stated that women have different patterns when the disease occurs. Women experience less obstructive coronary disease, and they present with more associated symptoms and higher rates of comorbidity (e.g., diabetes, hypertension, and previous heart failure) [10].

Although several studies have expounded these differences [6], only 50% of the trials have been conducted using a gender approach [7]. In fact, the guidelines for CVD prevention base their recommendations on studies conducted mainly among the male population [8]. As the burden of CVD is increasing in middle-aged women relative to men, a more profound understanding of the fundamental biological differences that exist between men and women is needed [11]. According to the report from the European Heart Health Strategy (EuroHeart) project [12], women are still underrepresented in many cardiovascular clinical trials, even though evidence shows significant sex differences in most areas of heart disease [8]. All of the aforementioned material has led to an increase in the interest in research regarding sex and gender differences in the appearance and development of CVD. To date, most of women’s specific risk factors have not been included in the guidelines for the prevention of CVD. In this sense, the European Guide on the prevention of cardiovascular disease in clinical practice [13] emphasizes the need to place “greater emphasis on an approach based on population and specific interventions of the disease and women” (p. 52). This guideline expounds that the determining factors in the development of the disease are not only genetic but also sociodemographic (e.g., sex, age, socioeconomic status, and educational level) and psychosocial.

With regard to psychosocial factors, previous studies have highlighted the influence of them in the development of CVD, showing sex differences in coping with stress, as well as in depression and anxiety disorders that are associated with a greater risk of CVD among women than among men [14]. The regulatory variables allow individuals to manage their emotions and behavior, which is of great importance in the promotion of healthy behaviors [15]. Thus, psychosocial factors also include emotional regulatory variables, making it necessary to intervene by promoting specific strategies to reduce depression, anxiety, and distress by using communicative and cognitive-behavioral strategies [16,17]. The WHO (World Health Organization) [18] warns that modifying one’s lifestyle could prevent more than three quarters of all deaths from CVD, along with an associated reduction in the economic costs. Chronic diseases, as in the case of CVD, entail some behavioral changes in the individuals who suffer from them [19]. If a gender approach is taken to understand differences between women and men in CVD, psychological interventions would be better suited to respond to these differences.

Concerning the psychological variables related to CVD, health perception is receiving increasing attention, with recent studies suggesting the link between self-reported health and cardiovascular health in CVD patients [20]. Moreover, perceptions of health are important to motivate people to change behaviors [14]. Taking into account this evidence, this study aimed to analyze differences between women and men patients with CVD in perceived health from a gender perspective, that is, by considering the role of sociodemographic, psychosocial, and emotional regulatory variables linked to this disease on their self-reported health status. Although there is an overlap between sex and gender constructs, in this study, we use the terms sex or sex differences when comparing the results based on a participant’s self-classification as male or female. Conversely, we use the term gender when accounting for the different implications of these sex differences in cardiovascular health. Thus, this study was designed acknowledging the greater focus of past research on sex differences and the need for a greater consideration of women’s health.

Therefore, the present research aims to address the aforementioned gap. Thus, we propose an investigation that responds to the need to explore the role of psychosocial variables to predict levels of perceived global health. Consistent with the literature reviewed, the main aim of the present research was to show, with a longitudinal study, the relationship between sociodemographic factors, emotional and regulatory variables, and perceived global health in a large sample of cardiovascular patients, all from a gender perspective. Firstly, the aim was directed to evaluate the differences between women and men in both sociodemographic and psychosocial variables, including emotional regulation strategies among a large sample of patients with CVD. Secondly, the aim was to focus on the analysis of the relationship between sociodemographic and emotional regulation variables with the level of perceived global health over time from a gender perspective.

### 1.1. Sociodemographic Variables Related to Cardiovascular Disease

Several studies underscore how sociodemographic variables, such as life situation (living alone or not) [21], socioeconomic status [22], and level of education [23,24], influence the development of CVD. In this sense, some studies warn of the following risk factors for CVD mortality: low socioeconomic status (low educational level, low income, low- work status, or residence in an area considered poor encompass characteristics that define a person with a low socioeconomic level), social isolation, and lack of social support [13,16,25,26]. Unfortunately, women seem to have a lower educational level [27], worse socio-economic status [22], live in a more isolated way [28], and experience an increased risk of developing CVD due to stress at mid-life compared to men [29].

### 1.2. Psychological Variables Associated with Cardiovascular Disease

Among the psychosocial risk factors, the European guide on the prevention of CVD in clinical practice [13] highlights aspects considered to be the greatest risk for experiencing a cardiovascular event. Some studies have shown the importance of the following psychological variables and emotional regulation strategies in cardiovascular health [30] as well as the existence of differences between women and men:

Health perception is being considered a relevant variable for cardiovascular health since recent research indicates its relationship with physical and psychological health status in clinical and non-clinical populations. For example, a 30-year follow-up study indicated that participants with ideal cardiovascular health reported higher scores on their perceived health in middle age [31]. Conversely, a lower standard of self-reported health was capable of predicting incident CVD in a 10-year follow-up of healthy adults [32]. Recent studies also suggest the association between positive affect and physical and psychological well-being [33], and the relationship of high positive affect and low negative affect with higher self-assessed health in cardiac patients [34]. Considering gender, a multi-country study indicates that women with ischemic heart disease show lower perceived mental and physical health than men [35].

Affective balance is one of the components that refers to subjective well-being and explains emotional responses when evaluating quality of life. It is the result of considering both the positive and negative emotions experienced over a period. Experiencing positive emotions is associated with a better health status in patients with CVD [36]. In this sense, Boehm and Kubzansky [37] explain that there is evidence that positive affective balance protects against CVD and is associated with slower progression of CVD. In contrast, it has been shown that negative affective balance is associated with a worse prognosis in cardiac patients [38]. Research on related constructs have shown that depression and anxiety disorders are associated with a greater risk of CVD among women compared with men [14] and that women with CVD reported less positive affect than men [39].

Regulatory negative affect self-efficacy seems to play a determining role as a predictor of cardiovascular health. Negative emotions are associated with a worse health status in patients with CVD [40]. Following this idea, Caprara and Steca [41] explain that self-efficacy in regulating negative emotions is positively related to life satisfaction and better adaptation to the disease. Therefore, it is necessary to analyze the self-regulation of negative affect and its effect on the perception of global health. Results from a non-clinical sample indicated that men show higher scores in self-efficacy to regulate negative emotions compared with women [42].

Anxiety and emotional regulation strategies seem to play a positive role in cardiovascular health. Recently, a systematic review and meta-analysis of randomized controlled trials of psychological interventions for CVD [30] showed that psychological interventions reduced cardiovascular mortality. Psychological interventions influence the levels of depressive, anxiety, and stress symptoms compared to control groups. In this sense, Chaves and Park [43] point out that changes in health behaviors depend on emotional regulation, and they conclude that those patients with greater positive affect and well-being present with changes in more positive, healthy behaviors over time. The patients with fewer psychological resources developed more negative behavioral changes over time. Chauvet-Gelinier and Bonin [40] highlight that cardiac rehabilitation seems to be a crucial step in improving patients’ outcomes by helping them to build strategies in order to manage daily stress.

Concerning the regulation of negative affect, and although research with clinical samples is scarce in this domain, research shows some sex differences in relation to depression and anxiety levels and the use of emotional regulation strategies [44]. There are diverse response modulation strategies differentiated by the type and level of activity required. Women, in comparison with men, have more of the stress-related behavioral profile that has been linked to CVD [45]. Women may be more vulnerable to the adverse effects of stressors on cardiovascular health, for example, by having a higher prevalence of psychosocial stressors [45]. In this sense, women reported engaging more frequently in emotional regulation strategies than men did, such as rumination which is associated with greater depression and anxiety [46]. Differences have been found in the approach coping strategy in patients with CVD; specifically, this strategy was inversely associated with CVD mortality in men but not in women [47]. The limited evidence in emotion regulation in CVD patients from a gender perspective indicates the need for more research in this area.

### 1.3. The Present Study

Based on previously reviewed studies and the objectives set, the first hypothesis (H1) was based on the relationship between sociodemographic and psychological variables with perceived health; thus, for example, it was expected that those patients who presented with better affective balance (high positive affect and low negative affect), higher confidence on their self-efficacy for emotional regulation, greater use of active strategies to regulate anxiety and less use of passive strategies would have a better health perception, both physical and mental. It was expected that these relationships would hold for both men and women. In this sense, a second hypothesis (H2) was based on the differences between men and women; it was expected that men would assert significantly higher scores in positive affect, self-efficacy for emotional regulation, active strategies, and better health perception (both physically and mentally) compared with women. Finally, given that this is a longitudinal study, we anticipated (H3) that the aforementioned relationships would be maintained over time with a gender approach, and therefore, the variables analyzed in the first phase would act as predictors of perceived health in the final phase.

## 2. Materials and Methods

### 2.1. Procedure and Study Design

This study is a prospective longitudinal one with patients with CVD who were part of the CORDIOPREV [48] study of the Reina Sofía University Hospital of Córdoba and Maimonides Biomedical Research Institute of Córdoba (IMIBIC), Spain. The study was approved by the Research Ethics Committee of the Servicio Andaluz de Salud and the Reina Sofía Hospital (30 June 2015). Participation in the study was totally anonymous and voluntary, and prior to its completion, participants were informed of the objectives of the research while their informed consent was collected. All measures were presented considering the differences between both sexes and a language inclusive for women and men. Data was collected at Time 1 (since April 2016) and Time 2 (since January 2017) in a clinical room of the Reina Sofía Hospital in Córdoba (Spain). For the current study, data was collected until June 2019. Each day, an average of eight cardiovascular patients came to the clinic and completed, in the presence of a member of the research team, a battery of online questionnaires created with the program Questback, version 10.9 (Unipark, Cologne, Germany). Following the main aim of the study, perceived global health was considered as an outcome variable while sociodemographic factors and emotional and regulatory variables were considered as predictors.

### 2.2. Study Participants

The sample consisted of a total of 747 patients at Time 1, of which 625 were men and 122 were women. At the time the data was analyzed the sample at Time 2 comprised 586 patients, of which 503 were men and 83 were women. All participants were patients involved in the CORDIOPREV study, who had an established coronary condition but had not suffered a clinical event in the last 6 months and had no other serious illness. An explanation of patient selection can be found on the project website (http://www.cordioprev.es/), indicating both the inclusion criteria (informed consent and diagnostic criteria) and the exclusion criteria (age, heart failure, ventricular dysfunction, serious risk factors, chronic diseases not related to coronary risk, or participants in other studies). Table 1 shows the sociodemographic characteristics of study participants by sex at both times (1 and 2).

### 2.3. Instruments

Sociodemographic variables: Data were collected to describe the age, sex, educational level, employment status, marital status and life together, and socioeconomic status.

Affective balance: In order to assess the affective state, two essential factors of emotional states, emotions of positive and negative characters, were measured to investigate the emotional instability from a short version of the Positive and Negative Affect Scale (PANAS) [49]. The adaptation of the PANAS to Spanish by López-Gómez, Hervás and Vázquez [50] was used. Participants responded to 12 items on a 7-point Likert scale, where 1 = strongly disagree and 7 = strongly agree, by first factor six negative items (e.g., “afraid”) and by second factor six positive items (e.g., “inspired”). The reliability was good (α = 0.80 and α = 0.89) in phase 1; (α = 0.84 and α = 0.89) in phase 2, respectively.

Regulatory negative affect self-efficacy: In order to assess how confident individuals were that they can manage their negative affect, the negative factor of the Spanish version of Regulatory Emotional Self-Efficacy Scale (RESE) [51] validated in that publication was used. Participants responded to the eight items (e.g., ”How confident are you that you can avoid flying off the handle when you get angry?”) on a 7-point Likert scale, where 1 = absolutely not capable and 7 = totally capable. The reliability of the scale in the original study was 0.82, and in this sample the reliability was also high (α = 0.88 and α = 0.90 in phase 1 and 2, respectively).

Stress and anxiety regulation strategies: In order to assess how individuals were able to manage their anxiety, the passive and physical strategy factors of the Anxiety Regulation Strategies Scale were used (STARTS, Authors Under Review). Participants responded to 18 items, answering the question: “What strategies do you use to control your stress or anxiety?” (e.g., 9 items for passive strategies: “Watching TV” and 9 items for physical strategies: “Going for a walk”) on a 7-point Likert scale, where 1 = never and 7 = always. Reliability was the same as in the original scale for both the passive strategy factor (α = 0.79 and α = 0.79, respectively) and the physical strategy factor (α = 0.73 and α = 0.75, respectively).

Perceived global health: In order to evaluate the perception of quality of life of the participants, the adaptation of the Spanish version of the Short Form 12 (SF-12) health survey, developed by Vilagut, Valderas, Ferrer, Garin, López-García, and Alonso [52], was used. Participants responded to 12 items to assess the degree of well-being and functional capacity of people, defining a positive and negative state of physical and mental health (e.g., “In general, how you would say your health is?”) on a Likert scale, which varies the number of choices depending on the item. Reliability was good (α = 0.81) and similar to the original Spanish scale, which ranged from 0.78 to 0.85.

### 2.4. Data Analysis

Statistical analyses were performed by using SPSS v.26 (IBM, Armonk, NY, USA). First, we carried out a descriptive analysis and comparison of means in both phases for the sociodemographic characteristics of the study participants according to their sex (H1). Second, the mean differences between males and females were tested and for bivariate correlations for all psychological variables studied at baseline and at follow-up phases, and a comparison was performed (H2). Finally, to explore the potential relationship between the sociodemographic variables and the psychological variables with perceived global health in phases 1 and 2, regression analyses were performed (H3).

## 3. Results

### 3.1. Differences by Sex in the Sociodemographic Variables Studied

To evaluate the sex differences in the sociodemographic variables studied, different chi-square tests were performed in phases 1 and 2 (Table 1). Also, age as a continuing variable was considered, and age differences were evaluated by a *t*-test, with no significant differences being found (t(43) = 42.02, M_male_ = 64.13, *SD* = 9.01; M_female_ = 66.61, *SD* = 8.82).

### 3.2. Psychological and Regulatory Strategies Variables Associated with Perceived Global Health in Cardiac Patients (H1)

Some preliminary analyses were conducted to test the relationship between the variables studied across both phases and for male and female samples. All relationships followed the expected direction. Therefore, men and women with a higher positive affect at baseline also showed a higher self-efficacy to regulate negative emotions and a better perception of health at baseline and in the follow-up measurement (in both phases). In contrast, males and females who experienced higher negative emotions and used more passive strategies (watch TV, medicine, infusion and bath) to cope with stressful situations at baseline significantly perceived worse health at the baseline and follow-up measurement (Table 2). A total of 161 patients did not participate in phase 2 (21.6%), and different ANOVAs did not show differences in the psychological variables evaluated in comparison with the other 586 patients who participated in both phases: positive affect (*F*(1,745) = 2.72, *p* = 0.10), negative affect (*F*(1,745) = 1.03, *p* = 0.31), affective balance (*F*(1,745) = 0.17, *p* = 0.68), regulatory negative affect self-efficacy (*F*(1,745) = 0.21, *p* = 0.64), and physical strategies (*F*(1,745) = 0.13, *p* = 0.72). However, ANOVAs showed significant differences between both samples in the use of passive strategies (*F*(1,745) = 4.28, *p* < 0.05), and health perceived (*F*(1,745) = 14.27, *p* < 0.001), where the patients who dropped-out of the study showed a higher use of passive strategies (*M* = 3.47; *SD* = 1.14) and worse health perceived (*M* = 2.87; *SD* = 0.65) compared to those who participated in both phases (*M* = 3.24; *SD* = 1.24; *M* = 3.06; *SD* = 0.55), respectively. Differences in health perceived remained significant for mental and physical health in similar directions (*F*(1,745) = 6.39, *p* < 0.01; *F*(1,745) = 17.51, *p* < 0.001, respectively).

### 3.3. Sex Differences in Psychological and Regulatory Strategies Variables and Perceived Global Health (H2)

Comparing the values of the variables in both sexes (Table 3), the results show that in phase 1 men presented significantly higher scores in positive affect, affective balance, and self-efficacy to regulate negative emotions, while women presented significantly higher scores in negative affect and the use of passive strategies to cope with stressful situations. In phase 2, only significant differences in greater affective balance remained for men and in a greater negative affect and the use of passive strategies for women. Finally, men showed significantly higher scores for perceived global health in both phases 1 and 2 compared with women.

### 3.4. Longitudinal Relationship of Sociodemographic Variables and Emotional and Regulatory Variables with Perceived Global Health (H3)

To explore the potential relationship of sociodemographic variables and emotional and regulatory variables with perceived global health in phases 1 and 2, regression analyses were performed. Levels of predictors in phase 1 were included to examine their potential predictive value on global, mental, and physical perceived health in phases 1 and 2. Also, two models were tested: one included only the most relevant sociodemographic characteristics for CVD (model 1), and the second model included these variables along with the psychological study variables (model 2). Table 4 shows the significant results for the entire sample, and Table 5 shows the results for each sex.

Results concerning the full sample showed that all models were significant (Table 4). However, the model 2 was able to explain a higher percentage of the total variance, reaching nearly 50% and more than one third of perceived mental health in phase 1 and phase 2, respectively. As for model 1, the age of participants showed predictive value on perceived mental health in phase 1, and of this variable and global perceived health in phase 2. Educational level was a significant predictor of the three perceived health variables tested in phase 1 and was able to predict physical health in phase 2. Concerning model 2, the age of participants was again a predictor of perceived mental health at both measurement moments and was a negatively significant predictor of perceived physical health in phase 1. As for the psychological variables, all of them resulted in being significant predictors of global, mental, and physical health both at phases 1 and 2. Also, their predictive value was as expected, with affective balance, self-efficacy for regulating negative emotions, and the use of physical strategies (walk, running, gym and gardening) to cope with stress being positively associated and the use of passive strategies having a negative value.

As regards results for women and men with CVD (Table 5), results also showed that all models were significant. However, as it was for the full sample, model 2 was able to explain a higher percentage of variance for all the health perceived variables. With respect to results concerning model 1, the socioeconomic level was a significant predictor of global and mental perceived health when data was divided by sex, being positively associated for men and negatively associated for women.

Considering the predictive value of model 2 in phase 2, results showed that sociodemographic characteristics were only relevant for male CVD patients, with the age of participants being a significant predictor of global and mental health, educational level being significant for mental health, and socioeconomic level for global and physical health. Most of psychological variables were again significant predictors of perceived global health for both sexes in an expected manner, with some exceptions. Specifically, the self-efficacy for regulating negative emotions predicted physical perceived health only for women, while the use of physical strategies for managing stress was a significant predictor of global and physical health only for men. As for the use of passive strategies, they were a significant predictor of global, mental, and physical perceived health for men, but only significant for perceived physical health for women.

## 4. Discussion

According to the research related to the influence of sociodemographic characteristics on cardiovascular health, and in line with past research, age and educational level are two of the variables related to health perception [23,24]. In the present study, cardiovascular patients showed significant differences by sex regarding sociodemographic characteristics. For men, age was positively related to health perception; however, for women, age had a negative relationship. Educational level had a significant relationship on the perceived global health of the female sample; in line with this result, Mackenbach et al. [27] also highlighted the relevance of educational level in improving cardiovascular health. So, female participants were distributed mostly between the categories “primary school” and “no school”, with a small percentage of women claiming to have higher education. Men in the category “no school” occupied a smaller percentage than women, being distributed among the categories that suppose a higher level of studies. Specifically, in the category of university studies, the percentage of men doubled compared to that of women. A low educational level has been associated with an increased incidence of coronary heart disease (CHD), mainly because of the strong inverse correlation of atherogenic risk factors and educational attainment. It is a consistent finding that men and women with lower education have a higher incidence of elevated blood pressure, total cholesterol, body mass index, and current smoking [53]. In relation to other notable differences in the sociodemographic characteristics of the present sample, the majority of participants confirmed that they were married. Some variations were observed in terms of sex differences. Among married people, the percentage of men was higher than that of women; however, among widowers, the number of women was much higher than that of men. Some investigations showed the association between marital status and CVD, outcomes, and cardiovascular risk factors [28,54].

Analysis of how certain psychological variables were related to cardiovascular health showed that men and women with a higher positive affect also demonstrated a higher self-efficacy to regulate negative emotions and a better perception of health, which coincides with the findings obtained by Sin et al. [36]. In contrast, men and women who experienced more negative emotions and used more passive strategies to cope with stressful situations perceived worse health at the baseline and follow-up measurement, as Meyer et al. [38] claim. Comparing the levels of the variables in both sexes, men presented significantly higher scores in positive affect, affective balance, and self-efficacy to regulate negative emotions, while women presented significantly higher scores in negative affect and the use of passive strategies to cope with stressful situations. These results coincide with research that highlights how an improvement in emotional regulation is positively related to psychological well-being and a better quality of life related to health [55], whereas inadequate stress management promoted the development of depressive symptoms [56], lower quality of life related to higher risk of physical dysfunction [57], and increase in heart failure [21].

As for the psychological variables, all of them proved to be significantly associated with perceived global health. Taking into account the predictive value of the regression models, the results showed that, among the sociodemographic characteristics, the age of participants stands out as a significant predictor of global and mental health. Educational level was significant for mental health, and socioeconomic level was significant for global and physical health. Differences exist between the sexes. For men, age was positively related to health perception. However, for women, it had a negative relationship. Educational level had a greater relationship on the perceived global health of women. Several studies have underscored how educational level and other sociodemographic characteristics [23,24] relate to the development of CVD.

Continuing with the psychological variables, they all showed the expected associations: positive affect, self-efficacy to regulate negative emotions, and the use of physical strategies to cope with stressful situations was related to greater perceived health. The use of passive strategies, however, was negatively associated with perceived health in the long term. Affective balance was the variable that had the greatest weight in explaining perceived global health. This result is especially relevant since some studies [36] explain that affective balance protects against CVD and is associated with a slower progression of the disease. Finally, it should be noted that self-efficacy to regulate negative emotions was positively related to perceived mental but not physical health. Previous studies have highlighted the relationship of psychosocial factors in the development of CVD, taking into account sex differences in coping with stress and emotions, as well as depression and anxiety disorders that are associated with a greater risk for CVD among women than among men [14]. Thus, despite emotional regulation strategies having a positive effect on health, differences between women and men in the use of regulatory strategies seem to have a different role on health promotion. In this sense, Chaves and Park [43] point out that changes in health behaviors depend on emotional regulation, and they conclude that those patients with greater positive affect and well-being present with changes in more positive, healthy behaviors over time, while patients with fewer psychological resources develop more negative behavioral changes over time.

One of the limitations of this study is the low representation of women, which coincides with the distribution of patients with CVD in other studies [6]. In future research, it would be better to have a more balanced number of women and men, since the disease is equally significant for both sexes. Another limitation is related to the drop-out of participants between both phases (21.6%). In this sense, Goldberg et al. [58] indicated possible differences in the characteristics of people who drop out of longitudinal studies, where cultural, psychological, and lifestyle behaviors or health variables could be implicated in rate participation. Despite the limitation of no data being available on those who did not participate in phase 2 of the study, the variables related to the higher use of passive strategies to cope with stressful situations and the worse perceptions of physical and mental health could act as indicators to avoid the dropouts. Finally, a last limitation is highlighted based on the directionality proposed in the regression analyses performed. Sociodemographic, emotional, and regulatory variables have been used as predictors of perceived health over time in the proposed regression models. It responds to dynamic theoretical models, in which cognitive and affective variables are interconnected. Because of that, and according to Cervone [59], the outcome variable (in our case “physical and mental health perception”) could be the result of dynamic transactions between the person and their environment. However, it is possible that some relationships analyzed would be bidirectional or going in the opposite direction. For example, the likelihood of engaging in some physical coping strategies (which were used as predictors) could be heavily influenced by one’s physical health perception (which was used as the outcome rather than a predictor variable).

## 5. Conclusions

In summary, based on the literature and the results obtained in samples of patients with CVD, we aimed to analyze which psychological variables are related to perceived health, from a gender perspective. Although the study of Raeisi-Giglou et al. [60] highlights the recent significant progress made in improving care, clinical decision-making, and policy implications for women with CVD, sex differences in both pathophysiology and biological risk factors could explain different prevalence rates, symptom profiles, and even medical outcomes; nonetheless, gender-specific distinctions with respect to psychosocial risk factors could explain this further regarding patients backgrounds [61].

The participants showed significant differences by sex in terms of sociodemographic characteristics. According to previous research [27,29,53,54], age is a variable that relates the perception of patients’ health, in a different way for women (negative relationship) and men (positive relationship). Another important sociodemographic variable is educational level, which had a significant relationship with the perceived global health of women. Regarding marital status, some research shows the association between marital status and CVD. Most of the sample answered that they were married.

Emotional regulation strategies are closely related to emotional well-being and perceived global health. However, among women, we found significant differences in the use of emotional regulation strategies to cope with stressful and adverse situations that occur on a daily basis, while men use more physical strategies and women use more passive strategies. These differences are maintained longitudinally and have a differential association with perceived global health. In this sense, the differences that we found between women and men indicate a relevant method of designing future brief interventions based on training in emotional regulation [62].

Our results support the need to create interventions as part of the programs for the rehabilitation and treatment of specific CVD for women, using a gender approach [7,8]. Interventions include psychosocial risk factors and disease and stress management programs, with consideration of age, educational level, social support, and specific psychosocial aspects of sex and gender, without trivializing psychosocial burdens and concerns. This analysis involves overcoming the biomedical conception of health, including research methodologies that incorporate a more contextual perspective of the health of women and men to identify the differences in the ways of getting sick for both sexes, as the differences that are due to gender are presented.

Based on our results, which are in line with the results of other authors, more and more research has revealed the differences between women and men to be considered in the intervention, highlighting the relevance of these interventions. Recently, a systematic review and meta-analysis of randomized controlled trials (RCTs) of psychological interventions for CVD [30] showed that psychological interventions reduce cardiovascular mortality. Another systematic review [63] also concluded that education-based interventions can improve CVD (including educational intervention along with exercise and psychological therapy). We found some group intervention programs designed to reduce stress in women with CVD, for example, the Stockholm Women’s Intervention Trial for Coronary Heart Disease, (SWITCHD) [64]. The desire to include women in research is not enough to establish it as gender-based research. What is crucial to the development of such research is the methodological focus and the underlying theory. A study in which sexual difference is a central analytical category and, as such, an explanation requires much more than the simple task of “adding” women to the data as a simple item of statistical information. We must realize the significance of being a woman or a man in health research [65,66]. In other words, it is necessary to include gender-specific strategies in the prevention of CVD through education and emotional regulation.

## Figures and Tables

**Table 1 jcm-09-03165-t001:** Demographic characteristics of study participants by sex.

			Phase 1						Phase 2				
			*n* = 747						*n* = 586				
Variables	Sex	%	Male	Female	X^2^	df	*p*	%	Male	Female	X^2^	df	*p*
			83.67%	16.33%					85.84%	14.16%			
Employment status (%)	Unemployed	7.9	6.4	15.6	14.10	3	0.003	6.8	4.4	8.4	149.47	4	<0.001
	Part-time work	3.3	3.2	4.1				2.9	3.0	2.4			
	Full-time work	19.3	20.5	13.1				16	22.1	8.4			
	Retired	69.5	69.9	67.2				58.3	70.0	49.4			
	Home care							16.3	0.6	31.3			
Partner (%)	Yes	89.2	91.8	75.4	28.50	1	0.001	82.4	91.3	73.5	22.48	1	<0.001
	No	10.8	8.2	24.6				17.6	8.7	26.5			
Marital status (%)	Single	3.9	3.7	4.9	56.35	5	0.001						
	Cohabiting partner	1.3	1.3	1.6									
	Married	84.5	87	71.3									
	Separated	2.0	2.1	1.6									
	Divorced	3.1	3.4	1.6									
	Widower	5.2	2.6	18.9									
Educational level (%)	No school	18.2	16.3	27.9	12.10	4	0.017						
	Primary school	49.1	49	49.2									
	Middle school	12.6	13.1	9.8									
	High school	9.9	10.4	7.4									
	University	10.2	11.1	5.7									
Economic level	<10,800 €	29.8	28.4	36.9	5.91	4	n.s.						
	10,800–22,000 €	41.9	42.2	40.2									
	22,000–43,000 €	15.2	16.1	10.7									
	>43,000 €	3.4	3.7	1.6									
	No answer	9.8	9.6	10.7									

**Table 2 jcm-09-03165-t002:** Bivariate correlations for all psychological and regulatory strategies variables studied at baseline and follow-up phases. Correlations for men are on the lower diagonal, and correlations for women are on the upper diagonal.

Variables	1	2	3	4	5	6	7	8	9	10	11	12	13	14	15	16	17	18
Phase 1. Base line																		
1. Positive Affect	-	−0.16	0.76 **	0.31 **	0.30 **	−0.15	0.53 **	0.59 **	0.35 **	0.57 **	−0.38 **	0.56 **	0.38 **	0.16	−0.22 *	0.59 **	0.62 **	0.43 **
2. Negative Affect	−0.23 **	-		−0.13	−0.02	0.14	−0.20 *	−0.24 **	−0.13	−0.10	0.42 **	−0.32 **	−0.34 **	0.25 *	0.39 **	−0.26 *	−0.33 **	−0.15
3. Affective Balance	0.79 **	−0.78 **	-	0.28 **	0.20 *	−0.19 *	0.47 **	0.55 **	0.31 **	0.45 **	−0.56 **	0.61 **	0.47 **	−0.07	−0.42 **	0.57 **	0.65 **	0.39 **
4. Reg. Negative Emotion Self-Efficacy	0.39 **	−0.28 **	0.42 **	-	0.13	−0.07	0.31 **	0.36 **	0.20 *	0.42 **	−0.40 **	0.47 **	0.35 **	0.17	0.01	0.44 **	0.42 **	0.32 **
5. Physical Strategies	0.07	0.12 **	−0.03	0.03	-	0.32 **	0.29 **	0.28 **	0.24 **	0.18	−0.15	0.19	0.11	0.41 **	0.21	0.10	0.10	0.08
6. Passive Strategies	−0.08	0.20 **	−0.17 **	−0.17 **	0.54 **	-	−0.24 **	−0.23 **	−0.21 **	−0.14	0.03	−0.09	−0.07	0.07	0.42 **	−0.29 **	−0.19	−0.25 *
7.Health Perceived	0.52 **	−0.34 **	0.54 **	0.40 **	−0.03	−0.35 **	-	0.86 **	0.92 **	0.50 **	−0.17	0.37 **	0.09	0.17	−0.17	0.68 **	0.46 **	0.61 **
8. Mental Health	0.54 **	−0.41 **	0.61 **	0.47 **	−0.05	−0.30 **	0.89 **	-	0.59 **	0.48 **	−0.29 *	0.45 **	0.17	0.08	−0.15	0.63 **	0.59 **	0.53 **
9. Physical Health	0.41 **	−0.21 **	0.39 **	0.27 **	−0.02	−0.32 **	0.93 **	0.65 **	-	0.37 **	−0.03	0.23 *	0.07	0.19	−0.14	0.51 **	0.28 *	0.56 **
Phase 2. Follow-up																		
10. Positive Affect	0.59 **	−0.19 **	0.49 **	0.35 **	0.03	−0.09 *	0.41 **	0.46 **	−0.29 **	-	−0.41 **	0.83 **	0.53 **	0.25 *	−0.09	0.57 **	0.59 **	0.42 **
11. Negative Affect	−0.28 **	0.46 **	−0.47 **	−0.30 **	0.07	0.15 **	−0.31 **	−0.40 **	−0.18 **	−0.36 **	-	−0.85 **	−0.46 **	−0.02	0.20	−0.45 **	−0.62 **	−0.22 *
12. Affective Balance	0.52 **	−0.40 **	0.58 **	0.39 **	−0.03	−0.15 **	0.43 **	0.52 **	0.28 **	0.82 **	−0.83 **	-	0.60 **	0.16	−0.17	0.60 **	0.72 **	0.38 **
13. Reg. Negative Emotion Self-Efficacy	0.34 **	−0.24 **	0.39 **	0.40 **	−0.06	−0.17 **	0.31 **	0.39 **	0.23 **	0.39 **	−0.35 **	0.48 **	-	0.08	−0.20	0.36 **	0.49 **	0.25 *
14. Physical Strategies	0.04	0.13 **	−0.06	−0.07	0.43 **	0.36 **	−0.11 *	−0.13 **	−0.06	0.06	0.12 **	0.04	−0.03	-	0.39 **	0.20	0.18	0.17
15. Passive Strategies	−0.07	0.18 **	−0.16 **	−0.18 **	0.30 **	0.52 **	−0.25 **	−0.26 **	−0.19 **	−0.08	0.19 **	−0.18 **	−0.18 **	0.64 **	-	−0.19	−0.13	−0.19
16. Health Perceived	0.46 **	−0.26 **	0.45 **	0.34 **	−0.01	−0.23 **	0.64 **	0.61 **	0.55 **	0.53 **	−0.44 **	0.59 **	0.49 **	−0.09	−0.30 **	-	0.83 **	0.91 **
17. Mental Health	0.47 **	−0.33 **	0.51 **	0.36 **	−0.05	−0.21 **	0.53 **	0.60 **	0.38 **	0.58 **	−0.53 **	0.67 **	0.53 **	−0.11 *	−0.30 **	0.89 **	-	0.52 **
18. Physical Health	0.37 **	−0.15 **	0.33 **	0.26 **	0.03	−0.19 **	0.61 **	0.51 **	0.59 **	0.41 **	−0.28 **	0.41 **	0.37 **	−0.05	−0.24 **	0.92 **	0.64 **	-

** *p* < 0.001; * *p* < 0.05.

**Table 3 jcm-09-03165-t003:** Psychological and regulatory strategies variables of study participants by sex in phase 1 and phase 2.

Variables		PHASE 1							PHASE 2						
	Sample	Total	Male	Female	*F*	*p*	η2	Potency	Total	Male	Female	*F*	*p*	η2	Potency
		*n* = 747	625	122	(1,745)				*n* = 586	503	83	(1,584)			
Positive Affect	*M*	5.01	5.06	4.76	6.25	0.01	0.008	0.704	5.12	5.14	4.97	1.57	0.21	0.003	0.24
	*SD*	1.23	1.22	1.26					1.16	1.14	1.3				
Negative Affect	*M*	2.66	2.59	3.01	12.24	0.001	0.016	0.938	2.39	2.33	2.72	7.23	0.007	0.012	0.766
	*SD*	1.22	1.21	1.26					1.21	1.18	1.36				
Affective Balance	*M*	2.35	2.47	1.75	14.73	0.001	0.02	0.97	2.73	2.81	2.25	5.78	0.017	0.01	0.67
	*SD*	1.92	1.9	1.91					1.96	1.91	2.23				
Reg. Negative Emotion Self-Efficacy	*M*	5.05	5.09	4.84	4.25	0.05	0.006	0.54	5.63	5.64	5.6	0.18	0.67	0	0.07
	*SD*	1.18	1.18	1.18					0.99	1	0.92				
Physical Strategies	*M*	3.07	3.08	3.02	0.39	0.53	0.001	0.1	2.91	2.9	3.02	0.79	0.38	0.001	0.14
	*SD*	1.03	1.02	1.11					1.11	1.12	1.1				
Passive Strategies	*M*	3.29	3.22	3.65	11.84	0.001	0.016	0.93	3.09	3.01	3.62	15.2	0.001	0.026	0.97
	*SD*	1.25	1.26	1.16					1.31	1.31	1.21				
SF12-Perceived Health	*M*	3.02	3.08	2.71	44.48	0.001	0.056	1	3.13	3.18	2.87	18.54	0.001	0.032	0.99
	*SD*	0.58	0.56	0.58					0.59	0.57	0.65				
SF12 -Mental	*M*	3.09	3.14	2.81	36.41	0.001	0.047	1	3.19	3.22	2.97	12.08	0.001	0.021	0.93
	*SD*	0.57	0.55	0.57					0.59	0.58	0.63				
SF12 -Physical	*M*	2.95	3.02	2.61	36.69	0.001	0.047	1	3.08	3.13	2.77	17.7	0.001	0.031	0.99
	*SD*	0.7	0.67	0.73					0.72	0.69	0.84				

*M* = Mean; *SD* = standard deviation; SF-12 = Short Form 12 health survey.

**Table 4 jcm-09-03165-t004:** Predictive variables of perceived health in phase 1 and phase 2. In the first regression model, sociodemographic variables are inserted as predictors of global, mental, and physical health in phase 1 and phase 2. In the second regression model, sociodemographic and psychological variables are inserted as predictors of global, mental, and physical health in phase 1 and phase 2.

Predictors of Perceived Health	Model 1			Model 2		
	SF12	SF12 MENTAL	SF12 PHYSICAL	SF12	SF12 MENTAL	SF12 PHYSICAL
Model SF12 Phase 1	*R*^2^_Adj._ = 0.02	*R*^2^_Adj._ = 0.02	*R*^2^_Adj._ = 0.04	*R*^2^_Adj._ = 0.42	*R*^2^_Adj._ = 0.48	*R*^2^_Adj._ = 0.28
	*F*(3,713) = 6.49 **	*F*(3,713) = 5.02 **	*F*(3,713) = 11.13 **	*F*(7,707) = 73.72 **	*F*(7,707) = 94.40 **	*F*(7,707) = 40.65 **
Model SF12 Phase 2	*R*^2^_Adj._ = 0.01	*R*^2^_Adj._ = 0.02	*R*^2^_Adj._ = 0.02	*R*^2^_Adj._ = 0.29	*R*^2^_Adj._ = 0.34	*R*^2^_Adj._ = 0.20
	*F*(3,557) = 3.71 *	*F*(3,557) = 4.50 **	*F*(3,557) = 5.57 **	*F*(7,538) = 32.38 **	*F*(7,538) = 40.90 **	*F*(7,538) = 20.78 **
Sociodemographic vs.	β Phase 1/2	β Phase 1/2	β Phase 1/2	β Phase 1/2	β Phase 1/2	β Phase 1/2
Age	0.03/0.10 *	0.12 **/0.16 **	−0.05/0.04	−0.02/0.05	0.06 */0.10 **	−0.08 */0.01
Educational level	0.15 **/0.08	0.10 */0.02	0.18 **/0.13 **	0.08 */0.00	0.01/−0.06	0.12 **/0.05
Economic level	0.04/0.08	0.04/0.02	0.03/0.09	0.02/0.07	0.02/0.02	0.02/0.09 *
Psychological vs.						
Affective balance				0.42 **/0.39 **	0.48 **/0.45 **	0.31 **/0.28 **
Regulation emotion self-efficacy				0.16 **/0.13 **	0.22 **/0.14 **	0.08 */0.11 *
Strategies regulatory physical				0.17 **/0.15 **	0.12 **/0.10 *	0.18 **/0.18 **
Strategies regulatory passive				−0.31 **/−0.21 **	−0.24 **/−0.16 **	−0.32 **/−0.23 **

β = Beta values, standardized coefficients in the linear regression model; ** *p* < 0.001; * *p* < 0.05; SF12 = Short Form 12 health survey; SF12 MENTAL = Short Form 12 mental health; SF12 PHYSICAL = Short Form 12 physical health; R2Adj. = percentage of variance explained.

**Table 5 jcm-09-03165-t005:** Predictive variables of perceived health in phase 1 and phase 2 in the male and female samples. In model 1, sociodemographic variables are inserted as predictors of global, mental, and physical health in phase 1 and phase 2. In model 2, sociodemographic and psychological variables are inserted as predictors of global, mental, and physical health in phase 1 and phase 2.

	MODEL 1			MODEL 2		
MALE SAMPLE	SF12	SF12 MENTAL	SF12 PHYSICAL	SF12	SF12 MENTAL	SF12 PHYSICAL
Model SF12 Phase 1	*R*^2^_Adj._ = 0.02	*R*^2^_Adj._= 0.03	*R*^2^_Adj._ = 0.03	*R*^2^_Adj._ = 0.41	*R*^2^_Adj._ = 0.48	*R*^2^_Adj._ = 0.26
	*F*(3,597) = 5.46 **	*F*(3, 597) = 7.44 **	*F*(3,597) = 6.52 **	*F*(7,591) = 59.31 **	*F*(7,591) = 79.90 **	*F*(7,591) = 31.17 **
Model SF12 Phase 2	*R*^2^_Adj._ = 0.02	*R*^2^_Adj._ = 0.03	*R*^2^_Adj._ = 0.02	*R*^2^_Adj._ = 0.27	*R*^2^_Adj._= 0.32	*R*^2^_Adj._ = 0.18
	*F*(3,480) = 4.88 *	*F*(3, 480) = 6.49 **	*F*(3,480) = 4.97 *	*F*(7,465) = 26.40 **	*F*(7,465) = 32.31 **	*F*(7,465) = 15.46 **
Sociodemographic vs.	β Phase 1/2	β Phase 1/2	β Phase 1/2	β Phase 1/2	β Phase 1/2	β Phase 1/2
Age	0.10 */0.15 *	0.18 **/0.19 **	0.01/0.09	0.02/0.09 *	0.09 */0.13 **	−0.04/0.05
Educational level	0.11 */0.01	0.03/−0.05	0.15 **/0.06	0.05/−0.03	−0.02/−0.08 *	0.10 */0.02
Economic level	0.10 */ 0.12 *	0.10 */0.06	0.07/0.14 *	0.04/0.09 *	0.05/0.03	0.03/0.12 **
Psychosocial vs.						
Affective balance				0.43 **/0.38 **	0.47 **/0.43 **	0.32 **/0.27 **
Regulation emotion self-efficacy				0.16 **/0.12 *	0.23 **/0.13 **	0.08/0.09
Strategies regulatory physical				0.14 **/0.15 *	0.09 */0.09 *	0.16 **/0.18 **
Strategies regulatory passive				−0.31 **/−0.21 **	−0.23 **/−0.16 **	−0.33 **/−0.21 **
**FEMALE SAMPLE**						
Model SF12 Phase 1	*R*^2^_Adj._ = 0.12	*R*^2^_Adj._= 0.09	*R*^2^_Adj._ = 0.12	*R*^2^_Adj._ = 0.37	*R*^2^_Adj._ = 0.41	*R*^2^_Adj._ = 0.24
	*F*(3,112) = 6.37 *	*F*(3,112) = 4.98 **	*F*(3,112) = 6.28 **	*F*(7,108) = 10.67 **	*F*(7,108) = 12.40 **	*F*(7,108) = 6.27 **
Model SF12 Phase 2	*R*^2^_Adj._ = 0.07	*R*^2^_Adj._ = 0.02	*R*^2^_Adj._ = 0.09	*R*^2^_Adj._ = 0.47	*R*^2^_Adj._ = 0.45	*R*^2^_Adj._ = 0.31
	*F*(3,73) = 2.90 *	*F*(3,73) = 1.47	*F*(3,73) = 3.61 *	*F*(7,65) = 10.07 **	*F*(7,65) = 9.32 **	*F*(7,65) = 5.62 **
Sociodemographic vs.	β Phase 1/2	β Phase 1/2	β Phase 1/2	β Phase 1/2	β Phase 1/2	β Phase 1/2
Age	−0.08/0.05	0.09/0.15	−0.19/−0.04	−0.12/−0.14	0.05/-0.03	−0.22 */-0.20
Educational level	0.33 **/0.34 *	0.34 **/0.24	0.26 **/0.34 *	0.19 */0.10	0.19 */0.03	0.16/0.13
Economic level	−0.20 */−0.16	−0.25 **/−0.19	−0.11/−0.11	−0.05/−0.01	−0.09/−0.03	−0.01/0.01
Psychosocial vs.						
Affective balance				0.31 **/0.48 **	0.41 **/0.55 **	0.18/0.31 **
Regulation emotion self-efficacy				0.13/0.29 *	0.16 */0.26 **	0.08/0.26 *
Strategies regulatory physical				0.25 */0.12	0.22 */0.10	0.23 */0.11
Strategies regulatory passive				-0.24 */-0.19	-0.18 */-0.08	-0.24 */-0.24 **

β = Beta values, standardized coefficients in the linear regression model; ** *p* < 0.001; * *p* < 0.05; SF12 = Short Form 12 health survey; SF12 MENTAL = Short Form 12 mental health; SF12 PHYSICAL = Short Form 12 physical health; R2Adj. = percentage of variance explained.

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
