# Peer review of "The Role of Emotional Regulation and Affective Balance on Health Perception in Cardiovascular Disease Patients According to Sex Differences"

_jcm, 2020, doi:10.3390/jcm9103165_

Round 1

Reviewer 1 Report

This manuscript aims at investigating CV patients' health perception putting attention on gender differences. The theme is certainly important as current literature needs a wider consideration of gender investigation in CV diseases' onset and progress. Gender differences also deserve greater attention in designing intervention aimed at contrasting CV diseases' evolution and their psychosocial impact. The longitudinal design of the study is also very valuable.

Below I report some notes and suggestions I hope may be useful in “rewriting” parts of the paper and in improving its impact on the literature.

First of all a substantial linguistic revision should be provided. There are several grammatical/semantic errors as well as redundancies and inaccuracies that authors need to address in order to increase the paper's readability (to give just an example: in the abstract 'it looks like it its necessary to include'; at pag. 2: 'women have been considered as being were protected" against CVD ').

I suggest to rethink the introduction clearly focusing on health perception and its value on patients' clinical and subjective state. Paper's title led the reader to believe that affective balance and health perception are the core outcomes of the study and that emotional regulation and sociodemographic variables are the hypothesized predictors. A few studies truly focused on patients' health perception are reported by authors and the reader discovers the role given to the variables only in the aim section. I strongly suggest to report a more focused literature as well as a deeper consideration of the role played by gender.

More details have to be reported as regards the cited studies. For instance, we read 'Although several studies have expounded these differences, only 50% of the trials have been conducted with consideration of gender differences' but there is no references about the cited trials  (Do they refer to national trials? Trials in Europe or what else?). Also the choice of predictors needs more justification. From current version of the paper it not clear the reason to choice self-efficacy in regulating negative affect. No studies is reported on  its investigation in the realm of cardiovascular diseases

Aims of the study should be better described and hypotheses should be clearly formulated on the bases of previous literature. A major attention has to be placed on the role of gender if authors really wanted to take 'a gender perspective'. In the current version  gender differences are investigated, and this is valuable, but this not enough to propose a gender perspective. Consideration of the appropriates of measures in terms of gender should be also added.

More information needs to be given on drop-out reasons and a comparison between people leaving the study and people staying in it should be performed.

The scale measuring the Stress and anxiety regulation strategies identifies two different strategies and the frequencies of their use. Also in this case no hypotheses are reported about the two strategies (and possible gender differences).

In the correlations table authors report results relative to negative and positive affect whereas on the subsequent regression analyses affective balance is used. I couldn't understand the reason of this choice.

As a final comment, part of the results (e.g., the different role of age for males and females) should be more discussed in the final section and more attention should be placed on the practical implications of the study findings.

Author Response

Response to Reviewer 1 Comments

This manuscript aims at investigating CV patients' health perception putting attention on gender differences. The theme is certainly important as current literature needs a wider consideration of gender investigation in CV diseases' onset and progress. Gender differences also deserve greater attention in designing intervention aimed at contrasting CV diseases' evolution and their psychosocial impact. The longitudinal design of the study is also very valuable.

Below I report some notes and suggestions I hope may be useful in “rewriting” parts of the paper and in improving its impact on the literature.

Point 1: First of all a substantial linguistic revision should be provided. There are several grammatical/semantic errors as well as redundancies and inaccuracies that authors need to address in order to increase the paper's readability (to give just an example: in the abstract 'it looks like it its necessary to include'; at pag. 2: 'women have been considered as being were protected" against CVD').

Response 1: A linguistic revision of English has been carried out by a specialized company (Proof Reading service.com). Following the suggestions of reviewer 1, the cited examples of the abstract and the introduction have been modified.

Point 2: I suggest to rethink the introduction clearly focusing on health perception and its value on patients' clinical and subjective state. Paper's title led the reader to believe that affective balance and health perception are the core outcomes of the study and that emotional regulation and sociodemographic variables are the hypothesized predictors. A few studies truly focused on patients' health perception are reported by authors and the reader discovers the role given to the variables only in the aim section. I strongly suggest to report a more focused literature as well as a deeper consideration of the role played by gender.

Response 2: a review has been made and new references have been added (for example: Gao, Z., Chen, Z., Sun, A., & Deng, X. (2019). Gender differences in cardiovascular disease. Medicine in Novel Technology and Devices4, 100025). This is a review that aims to provide an overview of the gender-related differences in various typical CVDs and to list and analyze the possible causes associated with the differences. This review emphasizes the need to take into account gender differences in determining the cardiovascular risk profile. This article helps us to review the literature on sex differences in CVD incidence risk factors.

Also, following the recommendations, this aspect has been linked to the variables of psychological adjustment and perception of health, introducing new references that affect how the perceptions of health are important to motivate people to change behaviors (for example: Lee, K. S., Feltner, F. J., Bailey, A. L., Lennie, T. A., Chung, M. L., Smalls, B. L., ... & Moser, D. K. (2019). The relationship between psychological states and health perception in individuals at risk for cardiovascular disease. Psychology research and behavior management, 12, 317.)

The following sentences has been added to the introduction: Concerning the psychological variables related to CVD, health perception is receiving increasing attention, with recent studies suggesting the link between self-reported health and cardiovascular health in CVD patients [18]. Moreover, perceptions of health are important to motivate people to change behaviors Previous studies have highlighted the influence of the psychosocial factors in the development of CVD, taking into account gender differences in coping with stress and emotions, as well as depression and anxiety disorders that are associated with a greater risk for CVD among women than among men [19].

And when the affective balance variable is mentioned: Health perception is being considered a relevant variable for cardiovascular health since recent research indicates its relationship with physical and psychological health status in clinical and non-clinical populations. For example, a 30-year follow-up study indicated that participants with ideal cardiovascular health reported higher scores on their perceived health in middle age [33]. Conversely, lower self-reported health was capable of predicting incident CVD in a 10-year follow-up of healthy adults [34]. Recent studies also suggest the association between positive affect and physical and psychological well-being [35], and the relationship of high positive affect and low negative affect with higher self-assessed health in cardiac patients [36]. Considering gender, a multi-country study indicates that women show lower perceived mental and physical health than men [37].

Point 3: More details have to be reported as regards the cited studies. For instance, we read 'Although several studies have expounded these differences, only 50% of the trials have been conducted with consideration of gender differences' but there is no references about the cited trials (Do they refer to national trials? Trials in Europe or what else?). Also the choice of predictors needs more justification. From current version of the paper it not clear the reason to choice self-efficacy in regulating negative affect. No studies is reported on its investigation in the realm of cardiovascular diseases

Response 3: research is cited from which the percentage of studies that take into account the differences between women and men is extracted: Maas, A. H. E. M. & Appelman, Y. E. A. Gender differences in coronary heart disease. Netherlands Heart J 2010, 18(12), 598-603. doi:10.1007/s12471-010-0841-y.

We studied variables related to the psychological well-being of patients with CVD, such as self-efficacy in regulating emotions, according to recent results that indicate the importance of positivity for healthier functioning in old age (Caprara et al., 2017) and its negative association with negative affectivity in patients with CVD (Steca et al., 2016) underscore the relevance of giving more consideration to this psychological factor. in future studies with this population.

Point 4: Aims of the study should be better described and hypotheses should be clearly formulated on the bases of previous literature. A major attention has to be placed on the role of gender if authors really wanted to take 'a gender perspective'. In the current version gender differences are investigated, and this is valuable, but this not enough to propose a gender perspective. Consideration of the appropriates of measures in terms of gender should be also added.

Response 4: Following the recommendations of reviewer 1, the hypotheses have been detailed at the end of the introduction and in the results presented. Consistent with the literature previously reviewed, the main aim of the present research is to show, with a longitudinal study the role of sociodemographic factors and emotional and regulatory variables on perceived global health in a large sample of cardiovascular patients all from a perspective based on gender differences. Firstly, the aim is directed to evaluate the sex differences in both sociodemographic and psychosocial variables, including emotional regulation strategies among a long sample of patients with CVD. Secondly, the aim is to focus on the analysis of the influence of sociodemographic and emotional regulation variables on the level of perceived global health over time from a gender approach. Based on previously reviewed studies and the objectives set, the first hypothesis (H1) was based on the relationship between sociodemographic and psychological variables with perceived health; thus, for example, it was expected that those patients who presented with better affective balance (high positive affect and low negative affect), higher confidence on their self-efficacy for emotional regulation, greater use of active strategies to regulate anxiety and less use of passive strategies would have a better health perception, both physical and mental. It was expected that these relationships would hold for both men and women. In this sense, a second hypothesis (H2) was based on the differences between men and women; it was expected that men would assert significantly higher scores in positive affect, self-efficacy for emotional regulation, active strategies and better health perception (both physically and mentally) compared with women. Finally, given that this is a longitudinal study, we anticipated (H3) that the aforementioned relationships will be maintained over time with a gender approach, and therefore, the variables analyzed in the first phase will act as predictors of perceived health in the final phase.

Also, sentence has been added in the procedure section: All measures were presented considering the differences between both sexes and a language inclusive for women and men.

Point 5: More information needs to be given on drop-out reasons and a comparison between people leaving the study and people staying in it should be performed.

Response 5: Following the recommendations of the reviewer 1, we have made the relevant analyzes and the following explanation has been included: A total of 161 patients did not participate in phase 2 (21.6%), and different ANOVAs did not show differences in the psychological variables evaluated in comparison with the other 586 patients who participated in both phases: positive affect (F(1,745) = 2.72, p = .10), negative affect (F(1,745) = 1.03, p = .31), affective balance (F(1,745) = 0.17, p = .68), regulatory negative affect self-efficacy (F(1,745) = 0.21, p = .64), and physical strategies (F(1,745) = 0.13, p = .72). However, ANOVAs showed significant differences between both samples in the use of passive strategies (F(1,745) = 4.28, p < .05), and health perceived (F(1,745) = 14.27, p < .001), where the patients who dropped-out of the study showed a higher number of passive strategies (M = 3.47; SD = 1.14) and worse health perceived (M = 2.87; SD = 0.65) compared to those who participated in both phases (M = 3.24; SD = 1.24; M = 3.06; SD = 0.55), respectively. Differences in health perceived remained significant for mental and physical health in similar directions (F(1,745) = 6.39, p < .01; F(1,745) = 17.51, p < .001, respectively).

In addition, it is added as a limitation: Another limitation is related with the drop-out of participants between both phases (21.6%). In this sense, Goldberg and his colleagues [56] indicated possible differences in the characteristics of persons who drop-out of longitudinal studies, where cultural, psychological, and lifestyle behaviors or health variables could be implicated in rate participation. Despite the limitation of no data being available on those who did not participate in phase 2 of the study, the variables related to the higher use of passive strategies to cope with stressful situations and the worse perceptions of physical and mental health could act as indicators to avoid the dropouts.

Point 6: The scale measuring the Stress and anxiety regulation strategies identifies two different strategies and the frequencies of their use. Also in this case no hypotheses are reported about the two strategies (and possible gender differences).

Response 6: In this paragraph, the inclusion of these variables is justified and that we have effectively incorporated it into the objective: Firstly, the aim is directed to evaluate the sex differences in both sociodemographic and psychosocial variables, including emotional regulation strategies among a long sample of patients with CVD.

In addition, the Hypothesis 2 is: It was expected that these relationships would hold for both men and women. In this sense, a second hypothesis (H2) was based on the differences between men and women; it was expected that men would assert significantly higher scores in positive affect, self-efficacy for emotional regulation, active strategies and better health perception (both physically and mentally) compared with women.

Point 7: In the correlations table authors report results relative to negative and positive affect whereas on the subsequent regression analyses affective balance is used. I couldn't understand the reason of this choice.

Response 7: In the correlation table, following the recommendations of reviewer 1, the results have been added on affective balance.

Point 8: As a final comment, part of the results (e.g., the different role of age for males and females) should be more discussed in the final section and more attention should be placed on the practical implications of the study findings.

Response 8: The following sentence is included in the final section: The participants showed significant differences by sex in terms of sociodemographic characteristics. According to the state of the art [28, 49-51], age is a variable that relates the perception of patients' health, in a different sense for women (negative relationship) and men (positive relationship). Another important sociodemographic variable is educational level, which has a significant relationship on the perceived global health of women. Regarding marital status, some research shows the association between marital status and CVD; most of the sample answered that they were married.

Based on our results, which are in line with the results of other authors, more and more research has revealed the differences between women and men to be considered in the intervention, highlighting the relevance of these interventions. Recently, a systematic review and meta-analysis of randomized controlled trials (RCTs) of psychological interventions for CVD [29] showed that psychological interventions reduce cardiovascular mortality. Another systematic review [60] also concluded that education-based interventions can improve CVD (including educational intervention along with exercise and psychological therapy). We found some group intervention programs designed to reduce stress in women with CVD, for example, the Stockholm Women's Intervention Trial for Coronary Heart Disease, (SWITCHD) [61].

Reviewer 2 Report

Re:    jcm-904222

Title: Influence of emotional regulation on affective balance and health perception in cardiovascular disease patients from a gender approach

Comments for Authors

This manuscript tackles a research question of clinical value, that is whether there are differences between men and women in their own psychological adjustment to CVD. However, there are various issues that warrant the authors’ attention. I hope my comments below will be helpful.

GENERAL COMMENT

  1. Some sentences are hard to understand; having a native English speaker reviewing the manuscript would be helpful. Moreover, some acronyms and Spanish words need to be spelled out/translated (e.g., in Table 3: RESE, Sumatorio, Pot.). Lastly, I recommend refraining from using causal language (e.g., X influences Y) given the observational nature of the current study and the considerable conceptual overlap between some variables (see point 7 below).

INTRODUCTION

  1. The Introduction reviews the literature on sex differences in risk factors for CVD incidence; yet, the current study aims are to evaluate psychological determinants of health perceptions in individuals who were already diagnosed with CVD. These two sections need to be better tied, for instance, by discussing psychological adjustment and health perceptions in the context of prevalent CVD in the Introduction.
  2. The title emphasizes potential gender differences, but “gender” and “sex” are used interchangeably throughout the manuscript even though they refer to distinct constructs. See https://orwh.od.nih.gov/sex-gender/sexgender-influences-health-and-disease/infographic-how-sexgender-influence-health

METHODS

  1. More information is needed to describe the current sample. For instance, what was the time since participants’ respective CVD diagnosis? Also, was a prognostic indicator available? These factors are important, particularly as they can impact psychological adjustment.
  2. Power appears limited among women (n=83) for the longitudinal analyses; estimates are likely unstable, which limits reliable conclusions. The small number of women should at least be acknowledged as a limitation in the Discussion.
  3. Alcohol intake is not a sociodemographic variable. Also, any reason why alcohol is listed in Table 1 but not included in the regression models?
  4. Several variables are highly conceptually correlated, which without surprise, will lead to statistically significant associations (e.g., physical regulation strategies such as going for a walk is likely to be related to perceived physical health; positive and negative affect is likely to be related to mental health). One could question whether the overlap is too important to try to distinguish some variables as “predictors” and others as “outcomes”.
  5. Should sex be included as a covariate in the analysis presented in Table 4?

RESULTS

  1. Because the mental and physical health subscales of the SF-12 are studied separately from the overall health score in the regression models, it would help interpretation to report these subscales as well in Tables 2 and 3.

DISCUSSION

  1. It seems premature to recommend interventions based on the current results, given that they are i) cross-sectional or longitudinal with less than one year apart, and ii) based on overlapping constructs, making it challenging to determine what is truly the predictor vs. the outcome, and as a result, which should be the intervention target.

Author Response

Response to Reviewer 2 Comments

This manuscript tackles a research question of clinical value, that is whether there are differences between men and women in their own psychological adjustment to CVD. However, there are various issues that warrant the authors’ attention. I hope my comments below will be helpful.

GENERAL COMMENT

  1. Some sentences are hard to understand; having a native English speaker reviewing the manuscript would be helpful. Moreover, some acronyms and Spanish words need to be spelled out/translated (e.g., in Table 3: RESE, Sumatorio, Pot.). Lastly, I recommend refraining from using causal language (e.g., X influences Y) given the observational nature of the current study and the considerable conceptual overlap between some variables (see point 7 below).

Response 1: A linguistic revision of English has been carried out by a specialized company (Proof Reading service.com). Following the suggestions of reviewer 2, the cited examples of the table 3 have been modified. The language has also been reformulated to avoid causality, avoiding ‘influences’ and showing relationships.

INTRODUCTION

  1. The Introduction reviews the literature on sex differences in risk factors for CVD incidence; yet, the current study aims are to evaluate psychological determinants of health perceptions in individuals who were already diagnosed with CVD. These two sections need to be better tied, for instance, by discussing psychological adjustment and health perceptions in the context of prevalent CVD in the Introduction.

Response 2: a review has been made and new references have been added (for example: Gao, Z., Chen, Z., Sun, A., & Deng, X. (2019). Gender differences in cardiovascular disease. Medicine in Novel Technology and Devices4, 100025). This is a review that aims to provide an overview of the gender-related differences in various typical CVDs and to list and analyze the possible causes associated with the differences. This review emphasizes the need to take into account gender differences in determining the cardiovascular risk profile. This article helps us to review the literature on sex differences in CVD incidence risk factors.

Also, following the recommendations, this aspect has been linked to the variables of psychological adjustment and perception of health, introducing new references that affect how the perceptions of health are important to motivate people to change behaviors (for example: Lee, K. S., Feltner, F. J., Bailey, A. L., Lennie, T. A., Chung, M. L., Smalls, B. L., ... & Moser, D. K. (2019). The relationship between psychological states and health perception in individuals at risk for cardiovascular disease. Psychology research and behavior management, 12, 317.)

The following sentences has been added to the introduction: Concerning the psychological variables related to CVD, health perception is receiving increasing attention, with recent studies suggesting the link between self-reported health and cardiovascular health in CVD patients [18]. Moreover, perceptions of health are important to motivate people to change behaviors Previous studies have highlighted the influence of the psychosocial factors in the development of CVD, taking into account gender differences in coping with stress and emotions, as well as depression and anxiety disorders that are associated with a greater risk for CVD among women than among men [19].

And when the affective balance variable is mentioned: Health perception is being considered a relevant variable for cardiovascular health since recent research indicates its relationship with physical and psychological health status in clinical and non-clinical populations. For example, a 30-year follow-up study indicated that participants with ideal cardiovascular health reported higher scores on their perceived health in middle age [33]. Conversely, lower self-reported health was capable of predicting incident CVD in a 10-year follow-up of healthy adults [34]. Recent studies also suggest the association between positive affect and physical and psychological well-being [35], and the relationship of high positive affect and low negative affect with higher self-assessed health in cardiac patients [36]. Considering gender, a multi-country study indicates that women show lower perceived mental and physical health than men [37].

  1. The title emphasizes potential gender differences, but “gender” and “sex” are used interchangeably throughout the manuscript even though they refer to distinct constructs. See https://orwh.od.nih.gov/sex-gender/sexgender-influences-health-and-disease/infographic-how-sexgender-influence-health

Response 3: The concepts sex and gender have been reviewed, marking the differences between both sexes in the data obtained and the influence of gender on these differences.

METHODS

  1. More information is needed to describe the current sample. For instance, what was the time since participants’ respective CVD diagnosis? Also, was a prognostic indicator available? These factors are important, particularly as they can impact psychological adjustment.

Response 4: The following sentences has been added to the “Study Participants”: All participants were patients involved in the CORDIOPREV study, who had an established coronary condition but had not suffered a clinical event in the last 6 months and had no other serious illness. An explanation of patient selection can be found on the project website (http://www.cordioprev.es/), indicating both the inclusion criteria (informed consent and diagnostic criteria) and the exclusion criteria (age, heart failure, ventricular dysfunction, serious risk factors, chronic diseases not related to coronary risk, or participants in other studies). Table 1 show the sociodemographic characteristics of study participants by sex at both times (1 and 2). Information has been added, through new analyzes, on the abandonment of participants between both phases and has been added as a limitation of the study.

  1. Power appears limited among women (n=83) for the longitudinal analyses; estimates are likely unstable, which limits reliable conclusions. The small number of women should at least be acknowledged as a limitation in the Discussion.

Response 5: The following sentence is added to the conclusions section: One of the limitations of this study is the low representation of women, which coincides with the distribution of patients with CVD in other studies [6]. In future research, it would be better to have a more balanced number of women and men, since the disease is equally important for both genders.

  1. Alcohol intake is not a sociodemographic variable. Also, any reason why alcohol is listed in Table 1 but not included in the regression models?

Response 6: Following the recommendations of the reviewer 2 and to avoid confusion, alcohol consumption is eliminated from the sociodemographic variables.

  1. Several variables are highly conceptually correlated, which without surprise, will lead to statistically significant associations (e.g., physical regulation strategies such as going for a walk is likely to be related to perceived physical health; positive and negative affect is likely to be related to mental health). One could question whether the overlap is too important to try to distinguish some variables as “predictors” and others as “outcomes”.

Response 7: indeed the psychological variables are related, based on a health model. In section 2.1. A study design section is included to specify the role of the variables: following the main aim of the study, perceived global health was considered as outcome variable while sociodemographic factors and emotional and regulatory variables were considered as predictors.

  1. Should sex be included as a covariate in the analysis presented in Table 4?

 Response 8: We have chosen to differentiate the weight of the sex variable in the regression equation to differentiate the samples. We believe this provides more information. Table 5 gives more exhaustive information on the difference between men and women.

RESULTS

  1. Because the mental and physical health subscales of the SF-12 are studied separately from the overall health score in the regression models, it would help interpretation to report these subscales as well in Tables 2 and 3.

Response 9: In the table 2 and 3, following the recommendations of reviewer 2, the results have been added on mental and physical health subscales of the SF-12.

DISCUSSION

  1. It seems premature to recommend interventions based on the current results, given that they are i) cross-sectional or longitudinal with less than one year apart, and ii) based on overlapping constructs, making it challenging to determine what is truly the predictor vs. the outcome, and as a result, which should be the intervention target.

Response 10: Based on our results, which are in line with the results of other authors, more and more research has revealed the differences between women and men to be considered in the intervention, highlighting the relevance of these interventions. Recently, a systematic review and meta-analysis of randomized controlled trials (RCTs) of psychological interventions for CVD [29] showed that psychological interventions reduce cardiovascular mortality. Another systematic review [60] also concluded that education-based interventions can improve CVD (including educational intervention along with exercise and psychological therapy). We found some group intervention programs designed to reduce stress in women with CVD, for example, the Stockholm Women's Intervention Trial for Coronary Heart Disease, (SWITCHD) [61].

Round 2

Reviewer 2 Report

Re:    jcm-904222_Revision 1

Title: The role of emotional regulation on affective balance and health perception in cardiovascular disease patients from a gender approach

Comments for Authors

The authors have been partly responsive to my first round of comments. Although their efforts are appreciated, I believe there are remaining points that should be addressed.

  1. Thank you for favoring non causal language throughout the manuscript; yet, some remain in the abstract, as well as in the study aims and discussion.

  2. Per my suggestion and the one of Reviewer 1, the authors added a paragraph on previous research about health perception to the Introduction. However, the Introduction remains scattered, with some paragraphs on sex differences in risk factors for CVD incidence in initially healthy populations, and other paragraphs on psychological sequelae of CVD in already ill populations. These are two distinct populations in which mechanisms at play are likely different. Because the latter topic is more closely tied to the aims and hypotheses of the current study, it should be the focus. As Reviewer 1 suggested, the flow of the Introduction would be improved by rethinking/rewriting this section with a clear focus on emotional factors and health perceptions among individuals who already have CVD, as well as on why these relationships should be evaluated with a gender/sex lens.
  1. Although the authors said they revised the use of “gender” and “sex” terms, they are still used interchangeably (e.g., first 3 sentences of section 1.3). While I recognize that there is substantial overlap between these two constructs, it is somewhat confusing to alternate between them throughout the article. I would recommend either to i) explicitly make the distinction between sex and gender in the Introduction and be specific about which construct has been used in prior studies and in the current one, or ii) acknowledge the overlap between these two constructs in the Introduction and inform the readers that, for the current study, you evaluated one of them and use that terminology for the rest of the paper.
  1. Thank you for stating explicitly which variables are considered outcomes and predictors, respectively, in the text. However, my previous concern was rather that many of these outcomes and predictors are highly conceptually correlated. For instance, using positive and negative affect as predictors is very closely related to mental health as an outcome. For this reason, it is not surprising to find statistically significant associations. Because of this conceptual overlap, it is also impossible to be sure which factors come first/are true predictors, even more so when relationships are assessed using a cross-sectional design or over a short follow-up period like <1 year. For instance, the likelihood of engaging in some physical coping strategies (used as a predictor here) could be highly influenced by one’s physical health, but physical health is used as an outcome rather than a predictor here. Thus, it is possible that for some relationships, it goes the other way around than what the authors initially speculated. This issue should be, at a minimum, mentioned as a limitation in the Discussion.

Author Response

Response to Reviewer 2 Comments

The authors have been partly responsive to my first round of comments. Although their efforts are appreciated, I believe there are remaining points that should be addressed.

Point 1: Thank you for favoring non causal language throughout the manuscript; yet, some remain in the abstract, as well as in the study aims and discussion. 

Impact

Response 1: Thank you for your suggestion. We have reviewed the entire manuscript again and indeed we have corrected some sentences in which terms of causality appeared in the text (prediction, impact, influence...). We believe there are none left. We have only kept the word "prediction" when talking about regression models in the results section or “impact” when we are reviewing previous studies published with this aim. [Lines: 199, 200, 204; 296-297, 341-345, 390, 425-427].

Point 2: Per my suggestion and the one of Reviewer 1, the authors added a paragraph on previous research about health perception to the Introduction. However, the Introduction remains scattered, with some paragraphs on sex differences in risk factors for CVD incidence in initially healthy populations, and other paragraphs on psychological sequelae of CVD in already ill populations. These are two distinct populations in which mechanisms at play are likely different. Because the latter topic is more closely tied to the aims and hypotheses of the current study, it should be the focus. As Reviewer 1 suggested, the flow of the Introduction would be improved by rethinking/rewriting this section with a clear focus on emotional factors and health perceptions among individuals who already have CVD, as well as on why these relationships should be evaluated with a gender/sex lens.

Response 2: The authors appreciate the reviewer suggestion. According to that, we have attempted to improve the flow of the introduction by highlighting the need of a gender approach in the study of psychosocial variables in a clinical population of patients with CVD (e.g., LINES: 89-91, 95-98, 105-107), and also, by including more previous evidence about the variables analyzed in the study and the gender differences found so far [e.g., Lines: 133, 143, 151-154, 170-172,193-196]. However, research in some topics is still limited in CVD population and only available in non-clinical samples [Lines: 170-172, 184-185]. We have made other changes to improve the coherence of introduction section; for example, we have change the order in the presentation of the variables, now starting with “health perception” as it is the outcome variable. We hope that these amendments have improved the clarity and focus of the introduction section.

Point 3: Although the authors said they revised the use of “gender” and “sex” terms, they are still used interchangeably (e.g., first 3 sentences of section 1.3). While I recognize that there is substantial overlap between these two constructs, it is somewhat confusing to alternate between them throughout the article. I would recommend either to i) explicitly make the distinction between sex and gender in the Introduction and be specific about which construct has been used in prior studies and in the current one, or ii) acknowledge the overlap between these two constructs in the Introduction and inform the readers that, for the current study, you evaluated one of them and use that terminology for the rest of the paper.

Response 3: Thank you for your comment, which has helped us to clarify further the aim of the manuscript. In response to this point, we have made the following changes:

  1. The use of the terms has been reviewed once more and some changes have been included to improve clarity: LINES: 71, 72, 114, 185, 201, 202, 504-505
  2. A paragraph has been included that attempts to integrate your both recommendations (i and ii). This paragraph is: “Taking into account this evidence, this study aimed to analyze differences between women and men patients with CVD in perceived health from a gender approach, that is, considering the role of sociodemographic, psychosocial and emotional regulatory variables linked to this disease on their self-reported health status. Although there is overlap between sex and gender constructs, in this study we use the terms sex or sex differences when comparing the results based on participant’s self-classification as male or female. Conversely, we use the terms gender or gender differences when accounting for the different implications of these sex differences in cardiovascular health. Thus, this study was designed acknowledging the greater focus of past research on sex compared to gender differences and the need for a greater consideration of women’s health”. Lines 95-107.

Point 4: Thank you for stating explicitly which variables are considered outcomes and predictors, respectively, in the text. However, my previous concern was rather that many of these outcomes and predictors are highly conceptually correlated. For instance, using positive and negative affect as predictors is very closely related to mental health as an outcome. For this reason, it is not surprising to find statistically significant associations. Because of this conceptual overlap, it is also impossible to be sure which factors come first/are true predictors, even more so when relationships are assessed using a cross-sectional design or over a short follow-up period like <1 year. For instance, the likelihood of engaging in some physical coping strategies (used as a predictor here) could be highly influenced by one’s physical health, but physical health is used as an outcome rather than a predictor here. Thus, it is possible that for some relationships, it goes the other way around than what the authors initially speculated. This issue should be, at a minimum, mentioned as a limitation in the Discussion.

Response 4:

Thank you again for your suggestion, according to it and following your suggestion we have added the next paragraph in the discussion section (Lines: 466-475):

“Finally, a last limitation is highlighted based on the directionality proposed in the regression analyses performed. Sociodemographic, emotional and regulatory variables have been used as predictors of perceived health over time in the proposed regression models. It responds to dynamic theoretical models, in which cognitive and affective variables are interconnected. Because of that, and according to Cervone (2005), the outcome variable (in our case " physical and mental health perception") could be the result of dynamic transactions between person-environment. However, it is possible that some relationships analyzed would be bidirectional or going in the opposite direction. For example, the likelihood of engaging in some physical coping strategies (which were used as predictors) could be heavily influenced by one's physical health perception (which was used as the outcome rather than a predictor variable).”

Reference: Cervone, D. (2005). Personality architecture: Within-person structures and processes. Annu. Rev. Psychol.56, 423-452.
